# MiR-182 Is Upregulated in Prostate Cancer and Contributes to Tumor Progression by Targeting MITF

**DOI:** 10.3390/ijms24031824

**Published:** 2023-01-17

**Authors:** M. Y. Cynthia Stafford, Declan J. McKenna

**Affiliations:** Genomic Medicine Research Group, Ulster University, Cromore Road, Coleraine BT52 1SA, UK

**Keywords:** prostate cancer, microRNA, miR-182, epithelial-to-mesenchymal transition, MITF, biomarker

## Abstract

Altered expression of microRNA-182-5p (miR-182) has been consistently linked with many cancers, but its specific role in prostate cancer remains unclear. In particular, its contribution to epithelial–to–mesenchymal transition (EMT) in this setting has not been well studied. Therefore, this paper profiles the expression of miR-182 in prostate cancer and investigates how it may contribute to progression of this disease. In vitro experiments on prostate cancer cell lines and in silico analyses of The Cancer Genome Atlas (TCGA) prostate adenocarcinoma (PRAD) datasets were performed. PCR revealed miR-182 expression was significantly increased in prostate cancer cell lines compared to normal prostate cells. Bioinformatic analysis of TCGA PRAD data similarly showed upregulation of miR-182 was significantly associated with prostate cancer and clinical markers of disease progression. Functional enrichment analysis confirmed a significant association of miR-182 and its target genes with EMT. The EMT-linked gene MITF (melanocyte inducing transcription factor) was subsequently shown to be a novel target of miR-182 in prostate cancer cells. Further TCGA analysis suggested miR-182 expression can be an indicator of patient outcomes and disease progression following therapy. In summary, this is the first study to report that miR-182 over-expression in prostate cancer may contribute to EMT by targeting MITF expression. We propose miR-182 as a potentially useful diagnostic and prognostic biomarker for prostate cancer and other malignancies.

## 1. Introduction

MicroRNAs (miRNAs) are small, non-coding RNA molecules that regulate gene expression by interacting with messenger RNAs (mRNAs). In prostate cancer, the aberrant expression of several miRNAs has been reported as a contributing factor to the development of this disease [1,2,3]. However, there is still much to be discovered about how specific miRNAs function within the various signaling pathways that control cell growth and behavior as prostate cancer progresses. For example, previous research in our laboratory has linked various miRNAs to cell cycle control, epigenetic regulation, and tumor hypoxia in prostate cancer [4,5,6], but their role in other mechanisms also needs explored.

One such mechanism is epithelial–mesenchymal transition (EMT), a reversible process whereby cells lose or suppress their epithelial phenotypes and gain mesenchymal phenotypes [7]. This change allows cancer cells to become more malignant, increasing their ability to migrate, invade, and metastasize. Several miRNAs have been shown to be involved in EMT in various cancer types, raising the suggestion that they may be suitable biomarkers for disease and possible targets for therapeutic intervention (reviewed in [7,8,9]). However, these reviews have also emphasized the need for more research to properly evaluate the various interactions of specific miRNAs in order for them to have any realistic clinical utility. Therefore, investigation into strategically selected miRNAs that can provide insight into the EMT process in cancer is needed. One miRNA that is worthy of further investigation is hsa-miR-182-5p (miR-182).

This is an interesting miRNA because the expression of miR-182 has been investigated in several cancers and cell types, but its function appears to change depending on the setting. It is generally reported to operate as an oncogene, with elevated expression observed in breast [10], head and neck squamous cell [11], hepatocellular [12], pancreatic [13], and bladder [14] carcinomas, among others. However, the EMT gene database, dbEMT2 [15], has classified the *MIR182* gene as having a dual role because it displays both oncogenic and tumor suppressive effects in different EMT studies. For example, while miR-182 has been found to promote EMT in several cancer cell types, conflicting results were observed in glioma [16,17,18], lung cancer [19,20], and colorectal cancer [21,22] cells. In the case of liver cancer cells, miR-182 over-expression had the effect of suppressing EMT [23]. In prostate cancer, evidence to date would suggest miR-182 acts in oncogenic fashion [24,25], but only two papers to our knowledge have explored a link to EMT in this disease, and the results were somewhat contradictory. One demonstrated how miR-182 over-expression impacted upon EMT-related pathways, but did not identify any specific target [26]. By contrast, the other study demonstrated down-regulation of miR-182 levels during EMT, but showed that its re-expression induced mesenchymal–to–epithelial transition (MET) by targeting SNAI2 [27]. Clearly, more study is needed to determine exactly how miR-182 function is linked to EMT in prostate cells, including identifying other targets it might have that influence malignant growth.

Therefore, in this study, we explored the expression of miR-182 in prostate cancer cells and tissue, then proceeded to identify a novel target of miR-182 linking it to EMT. We used this combined data to appraise its usefulness as a potential biomarker in prostate cancer.

## 2. Results and Discussion

### 2.1. Up-Regulation of miR-182 Expression Is Associated with Prostate Cancer

We used qRT-PCR to show that the expression of miR-182 in prostate cancer cell lines (DU145, 22RV1, and PC3) was significantly increased compared to normal prostate cell line RWPE-1 (Figure 1a). We then corroborated this result with analysis of the TCGA PRAD patient cohort, which also showed that the expression of miR-182 was significantly up-regulated in prostate cancer tumor tissue compared to normal prostate tissue (Figure 1b). Likewise, using separate GEO datasets, we also showed that serum expression of miR-182 was significantly increased in prostate cancer patients compared to non-cancerous control patients (Figure 1c). Analysis of TCGA PRAD Illumina 450 k methylation array data, spanning 7 CpG sites in the MIR182 gene promoter region, revealed that mean methylation levels were significantly reduced in tumor compared to normal tissue (Figure 1d). Functional enrichment analysis further revealed that miR-182, by virtue of targeting many cancer-related genes, was significantly associated with several gene set description terms related to prostate cancer (Table 1).

### 2.2. Higher Levels of miR-182 Are Associated with More Advanced Prostate Disease

Further UCSC Xena analysis of TCGA PRAD data demonstrated that higher miR-182 expression was significantly associated with clinicopathological markers of prostate cancer progression, including Gleason score, pathological T stage, and lymph node involvement (Figure 2). We wanted to explore the biological mechanisms for this. Additional functional enrichment analysis of miR-182 and its network of target genes showed a significant association with cellular processes related to EMT and TGF-β signaling, both of which are well-established factors in prostate cancer progression (Table 2 and Appendix A, Figure 3). Together, this proposes how up-regulation of miR-182 expression can subsequently exert a significant biological role in prostate cancer development and progression.

### 2.3. MITF Is a Novel Target of miR-182 in Prostate Cancer

We were interested in identifying a target of miR-182 that had not previously been demonstrated in prostate cancer. Given that miR-182 appears to have an oncogenic role in this setting and is involved with EMT, we sought to identify a target that was also related to EMT that would have detrimental effects if down-regulated by the action of miR-182 upon it. A systematic cross-referencing of gene lists from dbEMT, miRTarbase, and Regulome Explorer datasets revealed seven common genes as candidates for experimental validation in vitro (Figure 4).

We then performed PCR testing in vitro for the seven candidate targets in PC3 and DU145 cells (Appendix A). In pre-miR-182 transfected cells, the expression of MAP3K3, MITF, and SNAI2 expressions were consistently decreased, while ACTN4 and FOXO1 showed inconsistent results. CCND2 and PITPNM3 expression was undetectable in both cell lines. We, therefore, filtered the selection to focus on MAP3K3, MITF, and SNAI2, measuring the effect of both miR-182 over-expression and inhibition on their expression in normal and cancerous prostate cells. For MAP3K3 and SNAI2, miR-182 over-expression significantly reduced their expression, but inhibiting miR-182 had little effect (Appendix A). However, since SNAI2 had already been proven a target of miR-182 in prostate cancer, and there is little evidence for a MAP3K3 role in this disease, we were more interested in examining further the link between miR-182 and MITF. As its name suggests, melanocyte inducing transcription factor (MITF) is an important regulator of melanocyte function and plays a role in EMT associated with melanoma, where it appears to have a dual function dependent on the context [28]. However, it is also involved in a range of important biological processes, including proliferation, differentiation, invasion, and DNA repair, thus its aberrant expression has unsurprisingly been linked to other cancers, as well (reviewed in [29]). It has also been linked to EMT in some cancers [30,31], but not in prostate cancer. Likewise, MITF has only previously been investigated in vitro as a target of miR-182 in melanoma [32] and HEK293 [33] cells, but no study to date has validated a link between miR-182 and MITF in prostate cancer. Likewise, there are no published papers that have interrogated human patient cancer datasets for a correlation between miR-182 and MITF. We, therefore, considered it the best candidate of the seven genes to reveal novel findings and proceeded to investigate how its expression correlated with miR-182 in prostate cancer. We first performed a comprehensive series of in vitro transfection experiments to show that transient over-expression of miR-182 in both normal and cancerous prostate cells resulted in a significant reduction of MITF at mRNA levels, compared to control transfections (Figure 5a). Conversely, when we inhibited miR-182, we observed a significant increase of MITF expression in two of the cell lines (Figure 5b). This was encouraging because we know from our previous work that inhibiting miRNA activity does not always reveal an effect on the proposed target, since the knockdown must be almost complete, and any effect is often masked by the influence of other miRNAs that also target the mRNA under investigation. Over-expression of miRNA in cells by transfection gives more consistent results and was more relevant for this study, as abnormal up-regulation of miR-182 is associated with prostate cancer. We, therefore, used this approach to validate that MITF protein levels were also significantly reduced in each cell line when miR-182 levels were elevated (Figure 5c). We then corroborated this in vitro work with analyses of the TCGA PRAD dataset to confirm that miR-182 and MITF expression was significantly negatively correlated in prostate tissue (Figure 5d). Finally, analysis of TCGA PRAD data revealed that expression of MITF was significantly down-regulated in prostate cancer tumor tissue compared to normal prostate tissue (Figure 5e), which was the inverse of the miR-182 profile in these samples. MITF expression was also significantly lower in tumors with higher Gleason scores, but there was no significant correlation with pathological T-stage or N-stage (Appendix A).

Together, these data were consistent with the hypothesis that MITF is targeted by miR-182, demonstrating how aberrant up-regulation of miR-182 in prostate cells causes decreased expression of MITF. This is important because MITF plays an important role in both EMT and TGF-β signaling, as evidenced by functional enrichment analysis (Appendix A) and its immediate network of gene/protein interactions (Appendix A). However, it is important to remember that the relationship between miR-182 and MITF takes places against a larger network of interactions, which must be considered in evaluating their effect on cell behavior (Figure 6).

### 2.4. Potential of miR-182 as a Biomarker of Prostate Cancer

Given the correlation between miR-182 and various prostate cancer clinical parameters, measurement of miR-182 may have clinical utility as a diagnostic and/or prognostic biomarker for this disease. ROC curve analysis of the TCGA PRAD cohort demonstrates that miR-182 shows high potential for distinguishing between tumor and normal tissue (Figure 7a). Similarly, there is significant difference in miR-182 levels between groups of patients who show a different remission response after primary therapy, with higher levels showing poorer response (Figure 7b). However, there was no significant association between miR-182 expression and biochemical recurrence following therapy (Figure 7c). For survival analysis, the patient cohort was divided into quartiles based on their miR-182 expression levels. Kaplan–Meier graphs show that those in the highest quartile of miR-182 expression had significantly reduced Disease-Free Interval and Progression-Free Interval times, compared to those in the lowest quartile (Figure 7d,e). There was no significant difference between these quartiles for Overall Survival (Figure 7f), although that may be because the number of deaths in this cohort was low. Furthermore, since miR-182 over-expression has been consistently linked with many other cancers, it was no surprise to find high AUC values for miR-182 in multiple TCGA patient cohorts, suggesting the diagnostic potential of miR-182 in multiple cancer types (Appendix A).

Similarly, the expression levels of both miR-182 and MITF in tumor tissue is significantly correlated with survival outcomes in several other TCGA patient cohorts, indicating that they could be a useful biomarker for different cancers (Appendix A). This is particularly evidenced by the hazard ratios and highly significant *p*-values observed in kidney clear cell carcinoma and sarcoma (Appendix A).

### 2.5. Discussion

Although miR-182 has been studied in several cancers, its contribution to EMT in a prostate cancer setting had not been well studied, so we wanted to explore that relationship further in this study. This is the first report to present research showing how the up-regulation of miR-182 can contribute to prostate cancer development through its regulation of MITF levels.

We first established through cell line experiments, followed by analyses of TCGA and GEO prostate cancer datasets, that miR-182 over-expression is indeed associated with prostate cancer and progression of the disease (Figure 1, Table 1), in accordance with previous findings [24,25]. Interestingly, the increased miR-182 expression may be due to decreased DNA methylation in its promoter region. Others have shown miR-182 expression to be dependent on methylation status in ovarian [34] and esophageal [35] cancer, but this is the first report to consider epigenetic regulation of miR-182 in prostate cancer. Although it is highly methylated, previous work in our lab has shown that even a small reduction in methylation of promoter CpG sites can result in increased expression of the gene [5]. It is also worth highlighting that miR-182 expression can be profiled in serum samples from humans, because that lends itself utility as a circulating biomarker of prostate cancer. Measuring miR-182 levels in serum or urine would be less invasive than tissue biopsy profiling and could facilitate sequential bio-fluid sampling to monitor disease progression or response to treatment. Indeed, circulating levels of miR-182 have been successfully measured in lung [36], colorectal [37], breast [38], and gastric [39] cancer, suggesting its value as a potential biomarker of disease.

We also noted that significantly higher expression of miR-182 was associated with more advanced prostate disease, as measured by pathological grading and staging (Figure 2). This suggested that miR-182 may play a fundamental role in driving cancer progression, so we wanted to explore the possible biological pathways involved. Functional enrichment analysis confirmed miR-182 was significantly associated with EMT and TGF-β signaling (Table 2 and Appendix A, Figure 3), as others had found [16,17,18,19,20,21,22,23]. Notably, several of these include genes involved in pathways related to androgen receptor signaling, thereby implicating miR-182 in the androgen-regulated cistrome, as others have proposed [24,26,40]. Furthermore, Figure 3 shows a link between miR-182 and other pathways associated with aggressive prostate cancer, such as p53 and PI3K-Akt signaling.

To explore further the regulation of specific genes in prostate cancer, we wanted to identify an EMT-related target of miR-182 that had not previously been shown in this setting. By cross-referencing gene lists from three databases, we identified seven genes of interest (Figure 4). We performed in vitro cell transfection experiments to validate all these potential gene targets of miR-182 (Appendix A) and selected MITF as the most interesting one to investigate in more detail. We experimentally confirmed it as a target in vitro in four cultured prostate cell lines, demonstrating in particular that miR-182 over-expression significantly reduces MITF protein and mRNA levels (Figure 5). This is important, as miR-182 is up-regulated in prostate cancer. Incidentally, there are 12 validated isoforms of MITF protein, 4 of which are around the region of 59kDa, which explains the multiple bands visible in our Western blot results [41]. We corroborated this in vitro evidence with analysis of TCGA data to show a significant negative correlation between miR-182 and MITF, as expected if MITF was a target. MITF gene expression is also reduced in tumor tissue compared to normal tissue, which we propose is due to the concomitant up-regulation of miR-182 levels. This is significant because MITF is proposed to have a tumor suppressor role in other cancers [42,43]. In fact, aberrant expression or mutation of MITF is associated with disorders as varied as coloboma, osteopetrosis, microphthalmia, macrocephaly, albinism and deafness [44], Waardenburg syndrome [45], and Tietz syndrome [46]. This is because MITF is known to coordinate a wide range of biological process, thus its impact on cell function and in disease development is widespread and varied [29].

Less work has been carried out in prostate cancer, but one integrated study of online datasets and PC3 cell-line reported that MITF plays a tumor-suppressive role in this setting [47]. Taken together, this evidence illustrates that up-regulation of miR-182 could have a significant detrimental effect on prostate cell growth and function by targeting and down-regulating MITF expression. An overview of the interaction networks of both molecules demonstrates how altered expression of each will impact on many other genes and proteins, including those involved in EMT and TGF-β signaling (Figure 6, Appendix A, Appendix A). It is the holistic effect of all these interactions that will ultimately determine the effect of miR-182 upon prostate cancer development. Interestingly, MITF is associated with neuroendocrine differentiation in melanoma [48]; hence, it is tempting to speculate that it may perform similarly in prostate cancer, which would be another biological mechanism linking it to EMT and plasticity in prostate cells. Moreover, several miRNAs have been implicated in EMT-induced cellular plasticity in neuroendocrine prostate cancer [49]; thus, it would be interesting to explore the link between miR-182 and MITF in this context. Alternatively, it may be prudent to focus on disease sub-types using single-cell analysis and/or advanced proteomics to gain further insight into the combined biological function of MITF and miR-182. Prostate tumors are inherently heterogenous, which presents a problem for diagnosis and treatment of the disease [50]. It is important to realize that the overall tumor expression of MITF and miR-182 may be largely determined by certain cell types, such as immune cells or fibroblasts. This may shed further light on their function. For example, MITF may be important in the immune response, as it shows significant positive correlation with several markers of tumor-associated macrophages, which are known to feature prominently in prostate tumors (Appendix A) [51]. Identifying the precise pattern of their genomic and/or proteomic expression would help address the challenges that intra-tumor and inter-tumor heterogeneity present in the effective management of prostate cancer.

Given the consistent up-regulation of miR-182 in tumor tissues, we were interested in its potential as a clinically useful biomarker. Our data suggest that miR-182 expression profiling may indeed be useful as a diagnostic marker and may also help predict patient response to therapy (Figure 7). This corroborates previous work that has investigated the biomarker potential of miR-182 in prostate cancer (Appendix A). These studies are consistent in finding significant diagnostic value of miR-182 from patient plasma and tissue samples [52,53,54], although not in urine [55]. However, results for prognostic value are less conclusive, showing either no significance [54] or contrasting results [55,56]. Interestingly, one study noted a difference based on race when they found miR-182 was significantly predictive for biochemical recurrence in African Americans, but not in European Americans [53].

However, despite the data presented here and previously, it is unlikely that miR-182 will have sufficient specificity or sensitivity to be a useful disease biomarker on its own. A more likely scenario is that carefully selected miRNA expression profiles are combined with other biomarkers or more traditional pathological markers to help improve diagnostic or prognostic accuracy. These may include other genes or proteins in the miR-182/MITF network of interactions, especially if there is a focus on developing a panel focused on EMT. Previously, we have emphasized that multivariate panels are much more likely to have diagnostic and prognostic value in prostate cancer than single biomarkers [57,58], and we highlighted the importance of standardized approaches to clinical studies of miRNAs as biomarkers [59]. Furthermore, current models for prostate cancer risk prediction utilize various combinations of genomic, proteomic, and/or clinical measurements, including the Stockholm-3 risk-based model [60] and the European Randomised Study of Screening for Prostate Cancer risk calculator [61]. Our data suggest that miR-182 could be a useful addition to the list of variables to be included, either as a tissue or bio-fluid marker, as these models evolve to improve clinical decision making for prostate cancer patients. Of course, the utility of miR-182 as a potential biomarker can obviously apply to other malignancies, as well. Indeed, our analysis suggests it may be a particularly suitable predictor of survival and/or disease outcome in kidney renal clear cell carcinoma and sarcoma (Appendix A). Moreover, it is also worth noting that miR-182 is part of the highly conserved miR-183 family of microRNAs, which also includes miR-183 and miR-96. Others have suggested that profiling the miR-183 family, either individually or as a miR-182-96-183 cluster, is a promising strategy for development of biomarkers for several cancers, with the caveat that further research into these miRNAs is required to achieve this [62]. The findings presented in this study add further evidence and weight to this proposal.

## 3. Materials and Methods

### 3.1. Cell Culture and Transfections

Cell lines were purchased from American Type Culture Collection (ATCC, Rockville, MD, USA). Cells were authenticated by an in-house genotyping service, confirmed free of mycoplasma (InvivoGen, Toulouse, France), and used at low passage number (3 to 6). RWPE-1 is a non-cancerous prostate epithelial cell-line that was grown in keratinocyte growth medium, supplemented with 5 ng/mL human recombinant epidermal growth factor and 0.05 mg/mL bovine pituitary extract (Life Technologies, Paisley, UK). Human prostate cancer cell lines DU145, PC3, and 22RV1 were grown in RPMI-1640, supplemented with 10% fetal bovine serum and L-glutamine (Life Technologies). Cells were cultured at 37 °C, with a humidified atmosphere of 95% air and 5% CO_2_. For miRNA transfections, 100,000 cells were seeded per well in 6-well plates to ensure ~80% confluency at harvest. After 24 h, cells were transfected with miR-182 precursor (pre-miR-182; Assay ID PM12369), miR-182 inhibitor (anti-miR-182; Assay ID AM12369), or non-targeting negative control precursor (pre-miR-neg)(all ThermoFisher Scientific, Horsham, UK) at a final concentration of 25 nM using Lipofectamine 2000 (Life Technologies). After 48 or 72 h, cells were harvested for RNA or protein extraction. Pre-miR-182 (double-stranded RNA molecule designed to mimic endogenous mature miR-182) and anti-miR-182 (single stranded oligonucleotide designed to specifically bind and inhibit endogenous miR-182) are chemically modified molecules designed upon the mature miR-182-5p sequence UUUGGCAAUGGUAGAACUCACACU.

### 3.2. Quantitative Real-Time PCR (qRT-PCR)

Total RNA was extracted from cell lines using miRNeasy Tissue/Cells Advanced Mini Kit (Qiagen, Manchester, UK) according to the manufacturer’s instructions and RNA integrity confirmed by NanoDrop™ 2000 spectrophotometer (ThermoFisher Scientific, Waltham, MA, USA). A total of 500 ng RNA was used for first strand cDNA synthesis using random primers with Transcriptor high-fidelity cDNA synthesis kit (Roche, Sussex, UK) according to manufacturer’s instructions. Amplification of PCR products was quantified using FastStart SYBR Green Master (Roche) on a Roche LC480 Lightcycler, using primer sets for *MITF* (fw: GGGCTTGATGGATCCTGCTT, rv: GCTCTTGCTTCAGACTCTGTG) and housekeeping gene *ACTB* (fw: GGACTTCGAGCAAGAGATGG, rv: AGCACTGTGTTGGCGTACAG). Expression was normalized to *ACTB* and data generated from the combined results of at least four independent biological replicates.

qRT-PCR of miR-182 was performed using the miRCURY LNA^TM^ microRNA PCR system (Qiagen). A total of 20 ng template RNA was used in each first strand cDNA synthesis reaction. PCR was performed over 40 amplification cycles and fluorescence monitored on the Roche LC480 Lightcycler. Normalization was against housekeeping gene SNORD48. For all qRT-PCR miRNA analysis, data were generated from the combined results of at least three independent biological replicates.

### 3.3. Protein Analysis

Protein was extracted using Cell Lysis Buffer (Abcam, Cambridge, UK) with 2% *v*/*v* protease inhibitor (ThermoFisher Scientific). Western blots were performed using Bio-Rad mini-Protean TGX Gels and Trans-Blot^®^ Turbo Transfer System and reagents (Bio-Rad, Watford, UK). Antibodies used for blotting were rabbit-anti-MITF, with mouse-anti-GAPDH as loading control (both Proteintech, Manchester, UK). Membranes were blocked in 5% milk diluted in TBS-T (0.05%), followed by incubation in the appropriate secondary antibody (goat anti-rabbit IgG-HRP (1:5000) or goat anti-mouse IgG-HRP (1:5000), both Proteintech). Luminescence was revealed by incubation with enhanced chemiluminescent reagent (ThermoFisher Scientific) and signal detected on a G:BOX F3 imaging system (Syngene, Cambridge, UK). At least four biological replicates per experiment were conducted.

### 3.4. Databases

The Cancer Genome Atlas (TCGA) Prostate Adenocarcinoma (PRAD) repository data were accessed at http://portal.gdc.cancer.gov/projects (accessed on 14 January 2022). Analysis was performed using The University of California Santa Cruz Xena Functional Genomics Explorer (UCSC Xena) (http://xenabrowser.net/, accessed on 26 July 2022) [63] and CancerMIRNome (http://bioinfo.jialab-ucr.org/CancerMIRNome/, accessed on 28 July 2022) [64] analysis tools. Serum miR-182 expression data were analyzed from Gene Expression Omnibus (GEO) (http://www.ncbi.nlm.nih.gov/geo/, accessed on 10 May 2022) datasets GSE112264 [65] and GSE113486 [66]. Functional enrichment analysis was performed using clusterProfiler [67] in CancerMIRNome and the Database for Annotation, Visualization and Integrated Discovery (DAVID)(http://david.ncifcrf.gov/home.jsp, accessed on 21 July 2022) [68,69]. Targets of miR-182 were identified using miRTarBase (http://mirtarbase.cuhk.edu.cn/, accessed on 14 February 2021) [70], the EMT gene database dbEMT2 (http://dbemt.bioinfo-minzhao.org/, accessed on 16 February 2021) [15], and Regulome Explorer (http://explorer.cancerregulome.org/, accessed on 15 February 2021), which contains primary prostate cancer data from a single study [71]. The protein–protein interaction (PPI) network of MITF was generated in STRING (https://string-db.org/, accessed on 24 March 2022) [72], followed by functional enrichment analysis using Kyoto Encyclopedia of Genes and Genomes (KEGG) annotation. Additional survival analysis was performed using Kaplan–Meier Plotter (KM-Plotter) (http://kmplot.com/analysis/, accessed on 25 March 2022) [73]. Network analyses were performed and visualized using GeneMANIA (https://genemania.org/, accessed on 21 April 2022) [74] and miRTargetLink 2.0 (http://ccb-compute.cs.uni-saarland.de/mirtargetlink2, accessed on 22 April 2022) [75].

### 3.5. Statistics

Graphs were generated using Graphpad PRISM v6. Unless otherwise stated, all bar graphs show mean ± standard error of at least three biological replicates, with statistical significance assessed by paired *t*-test. All boxplots show mean and Tukey whiskers, with statistical significance assessed by either unpaired *t*-test with Welch’s correction or non-parametric Kruskal–Wallis one-way ANOVA with Dunn’s Multiple Comparison Test. Statistical significance for scatterplots was assessed by Pearson’s correlation with *p*-values adjusted for multiple hypothesis testing. Statistical significance for Kaplan–Meier graphs was assessed by log-rank (Mantel–Cox) test. For multiple hypothesis correction in functional enrichment tables, the adjusted *p*-value used Benjamini and Hochberg procedure. For hazard ratio (HR), KM-Plotter utilized Cox proportional analysis with auto-selected cut-off. For all analyses, data were considered significant where * *p* < 0.05, ** *p* < 0.01, *** *p* < 0.001.

## 4. Conclusions

We have shown that miR-182 is significantly upregulated in prostate cells, as well as demonstrated that high levels of miR-182 expression are associated with clinicopathological markers of prostate cancer progression. This is the first study to show that miR-182 targets MITF in prostate cells, and we propose this is one mechanism by which it can influence the process of EMT in the progression of this disease. Further work is needed to examine this relationship further, but we nonetheless propose that miR-182 shows considerable promise as a diagnostic or prognostic marker for prostate cancer and other malignancies.

## Figures and Tables

**Figure 1 ijms-24-01824-f001:**
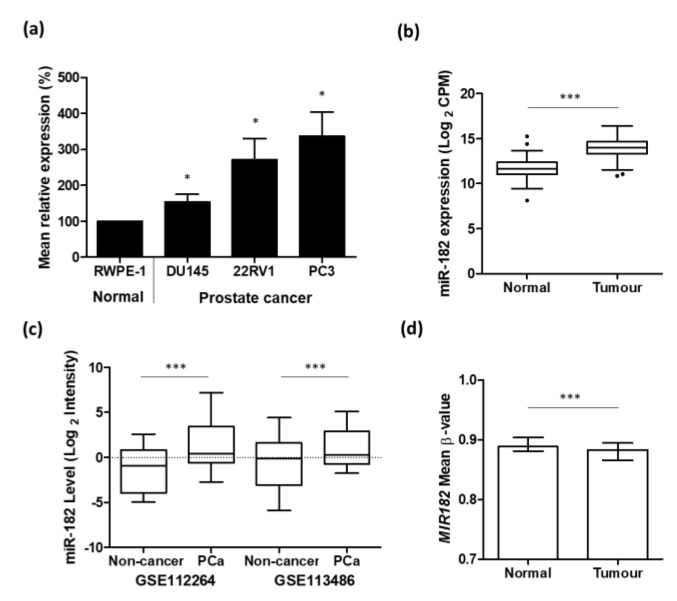
Up-regulation of miR-182 is associated with prostate cancer. (**a**) qRT-PCR shows relative mean miR-182 expression in DU145, 22RV1, and PC3 prostate cancer cell lines is significantly higher than that in normal prostate cell-line, RWPE-1 (*n* = 4, housekeeping: Snord48) (unpaired *t*-test, * *p* < 0.05). (**b**) UCSC Xena analysis of TCGA PRAD samples shows miR-182 expression is significantly increased in prostate tumor tissue (*n* = 494) compared to normal prostate tissue (*n* = 52). (Welch’s *t*-test *** *p* < 0.001). (**c**) miR-182 is significantly elevated in the serum of prostate cancer patients compared to healthy, non-cancer control patients. Data from GEO datasets GSE112264 (*n*, Non-cancer = 41, PCa = 809) and GSE113486 (*n*, Non-cancer = 100, PCa = 40). (One-way ANOVA with multiple comparison tests, *** *p* < 0.001). (**d**) Mean β-value methylation of 7 CpG sites in *MIR182* promoter region is significantly reduced in prostate tumor tissue (*n* = 431) compared with normal prostate tissue (*n* = 33). Wilcoxon paired test, *** *p* < 0.001. Graph generated from UCSC Xena analysis of TCGA PRAD Illumina 450 K methylation array data. Bars show median and interquartile range. CPM = Counts per million; *n* = number; PCa = prostate cancer.

**Figure 2 ijms-24-01824-f002:**
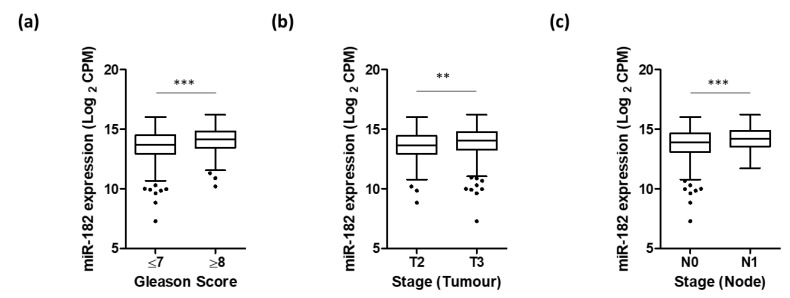
Higher expression of miR-182 is associated with more advanced prostate disease. UCSC Xena analysis of TCGA PRAD samples shows expression of miR-182 is significantly higher in patients with (**a**) Gleason score ≥ 8 (*n* = 211) compared to those scored ≤ 7 (*n* = 334), (**b**) pathological stage T3 (*n* = 310) compared to T2 (*n* = 215), and (**c**) pathological stage N1 (*n* = 81) compared to N0 (*n* = 368). (All Welch’s *t*-test, ** *p* < 0.01, *** *p* < 0.001). *n* = number.

**Figure 3 ijms-24-01824-f003:**
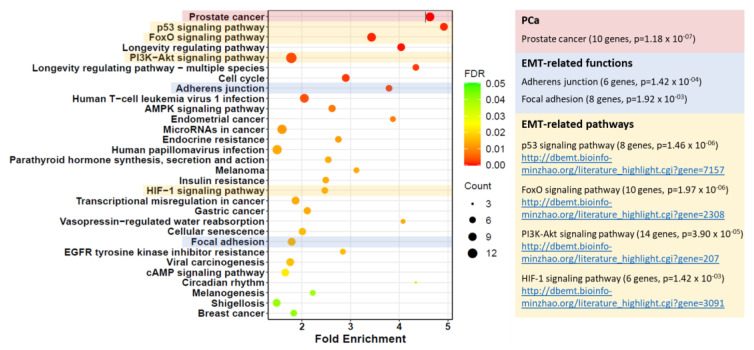
Bubble plot of KEGG functional enrichment analysis of miR-182-5p targets generated by CancerMIRNome. “Prostate cancer” (PCa) was the top enriched pathway (10 genes, *p* = 1.18 × 10^−7^). The EMT-related functions, “Adherens junction” and “Focal adhesion”, as well as EMT-related signalling pathways “p53” (https://dbemt.bioinfo-minzhao.org/literature_highlight.cgi?gene=7157), “FoxO” (https://dbemt.bioinfo-minzhao.org/literature_highlight.cgi?gene=2308), “PI3K-Akt” (https://dbemt.bioinfo-minzhao.org/literature_highlight.cgi?gene=207), and “HIF-1” (https://dbemt.bioinfo-minzhao.org/literature_highlight.cgi?gene=3091) were also significantly enriched (*p* < 0.05, URLs: dbEMT2 literature links).

**Figure 4 ijms-24-01824-f004:**
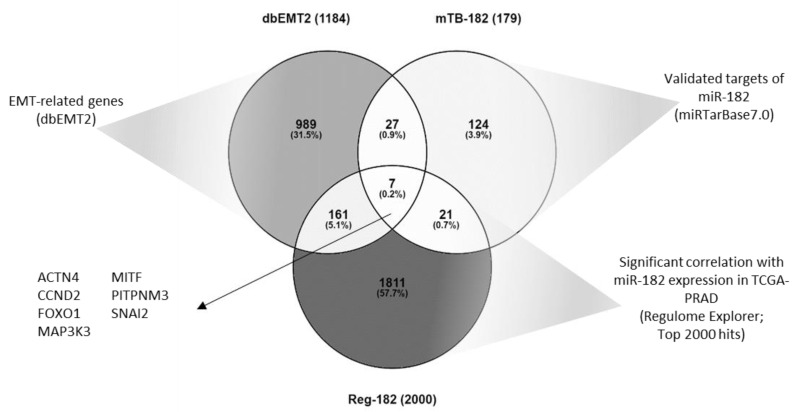
Seven genes common to datasets from dbEMT2, miRTarBase7.0, and Regulome Explorer TCGA-PRAD databases were identified as potential targets of miR-182.

**Figure 5 ijms-24-01824-f005:**
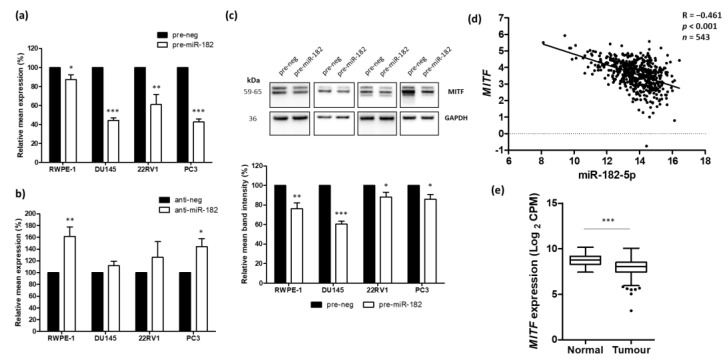
Validation of MITF as a novel target of miR-182 in prostate cancer cells. (**a**) RT-qPCR (n = 4) shows over-expression of miR-182 causes significant down-regulation of MITF in normal and cancerous prostate cell lines (paired *t*-test, * *p* < 0.05, ** *p* < 0.01, *** *p* < 0.001). (**b**) RT-qPCR (n ≥ 3) shows inhibition of miR-182 causes significant up-regulation of MITF in RWPE-1 and PC3 cells (paired *t*-test, * *p* < 0.05, ** *p* < 0.01). (**c**) Representative and quantified Western blotting (n ≥ 4) shows over-expression of miR-182 causes significant down-regulation of MITF in normal and cancerous prostate cell lines (paired *t*-test, * *p* < 0.05, ** *p* < 0.01, *** *p* < 0.001). (**d**) CancerMIRNome analysis of TCGA prostate specimens, including normal (n = 52) and tumor (n = 491) tissue samples, shows the expression of miR-182 and MITF are significantly negatively correlated (Pearson correlation, *p* < 0.001). (**e**) UCSC Xena analysis of TCGA-PRAD samples shows MITF expression is the inverse of miR-182, being significantly reduced in tumor (n = 497) tissues relative to normal (n = 52) tissue (Welch’s *t*-test, *** *p* < 0.001).

**Figure 6 ijms-24-01824-f006:**
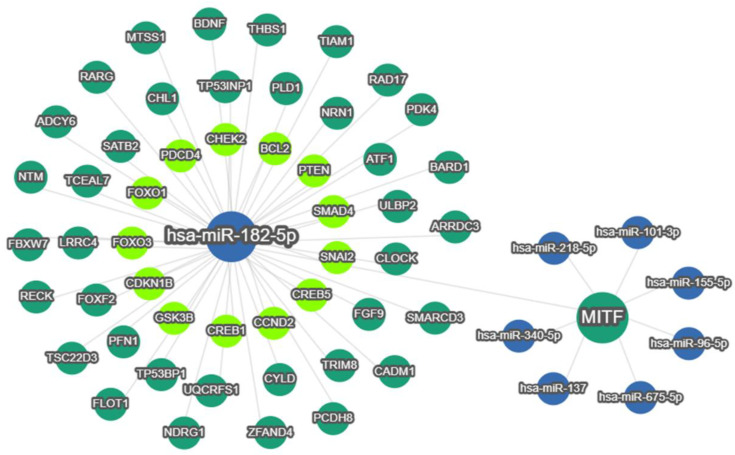
miRTargetLink2.0 visualisation of miR-182-5p and MITF bidirectional interactions. miR-182-5p and MITF were input items, while the connected nodes were gene/miRNA interactions validated by strong experimental evidence (qRT-PCR, Western blot and Luciferase reporter assay). Blue nodes: miRNAs; Green nodes: genes; Bright green nodes: genes significantly associated with prostate cancer and/or EMT.

**Figure 7 ijms-24-01824-f007:**
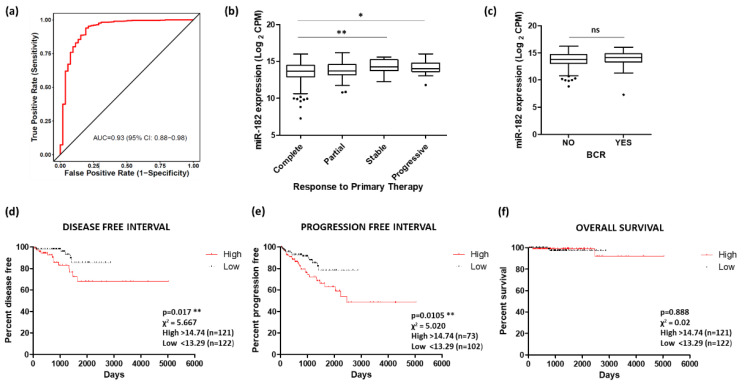
Potential of miR-182 as a biomarker of prostate cancer. (**a**) ROC curve analysis demonstrating that miR-182 shows high potential for distinguishing between tumor and normal tissue. Analysis performed using CancerMIRNome based on TCGA PRAD patient cohort. (**b**) Significant difference in miR-182 levels between patient remission response after primary therapy (*n*, complete = 380, partial = 41, stable = 27, progressive = 31). (One-way ANOVA with multiple comparison tests, * *p* < 0.05, ** *p* < 0.01). (**c**) Biochemical recurrence (BCR) showed no significant difference with levels of miR-182 (*n*, no recurrence = 407, recurrence = 61). (Welch’s *t*-test, *p* = ns). For KM plots, patients were divided into quartiles based on miR-182 expression. Quartile with highest miR-182 expression showed significantly reduced time for (**d**) Disease-free interval and (**e**) Progression-free interval, compared to quartile with lowest miR-182 expression. (**f**) No significant difference was found between these quartiles for Overall survival. (All log-rank (Mantel–Cox) test, ** *p* < 0.01). Data analysis for (**b**)–(**f**) was performed using UCSC Xena based on TCGA PRAD patient cohort. *n* = number; ns = non-significant.

**Table 1 ijms-24-01824-t001:** Functional enrichment analysis of miR-182 in prostate cancer. Table shows the significant association of miR-182 target genes with Gene Set descriptions related to prostate cancer. Analysis performed using clusterProfiler in CancerMIRNome.

Knowledge Base	ID	Description	Count/Total	Adjusted *p*-Value ^1^	Target Gene Symbol
KEGG	hsa05215	Prostate cancer	10/179	2.29 × 10^−5^	CDKN1A; FOXO1; EP300; CREB1; BCL2; IGF1R; PTEN; GSK3B; CREB5; CDKN1B
	hsa05206	MicroRNAs in cancer	11/179	1.15 × 10^−2^	CDKN1A; EP300; BCL2; CCND2; PDCD4; RECK; PTEN; NOTCH2; CDKN1B; THBS1; TRIM71
DO	DOID:10283	Prostate cancer	14/179	1.26 × 10^−2^	CDKN1A; BCL2; CCND2; SNAI2; SMAD4; IGF1R; BRIP1; BAG1; PTEN; CHEK2; CDKN1B; NDRG1; TNFRSF10A; NPM1
DisGeNET	umls:C1654637	Androgen independent prostate cancer	7/179	1.64 × 10^−2^	CDKN1A; FOXO3; BCL2; PTEN; CDKN1B; NR3C1; RGS2
	umls:C2931456	Prostate cancer, familial	4/179	2.20 × 10^−2^	PTEN; CHEK2; CDKN1B; RASA2

KEGG = Kyoto Encyclopedia of Genes and Genomes; DO = Disease Ontology. ^1^ Adjusted *p*-value for multiple hypothesis correction used Benjamini and Hochberg procedure.

**Table 2 ijms-24-01824-t002:** Functional enrichment analysis of miR-182 related to EMT. Table shows the significant association of miR-182 target genes with Gene Set descriptions related to EMT. Analysis performed using clusterProfiler in CancerMIRNome.

Knowledge Base	ID	Description	Count/Total	Adjusted *p*-Value ^1^	Target Gene Symbol
KEGG	hsa04520	Adherens junction	6/179	3.45 × 10^−3^	EP300; SNAI2; SMAD4; IGF1R; ACTN4; CTNNA3
	hsa04510	Focal adhesion	8/179	1.58 × 10^−2^	BCL2; CCND2; IGF1R; ACTN4; PTEN; GSK3B; THBS1; PPP1R12A
GO-BP	GO:0060485	Mesenchyme development	11/179	4.10 × 10^−3^	FGF9; BCL2; PDCD4; SNAI2; SMAD4; FOXF2; PTEN; GSK3B; CITED2; TIAM1; ZFP36L1
	GO:0001837	Epithelial to mesenchymal transition	7/179	8.31× 10^−3^	PDCD4; SNAI2; SMAD4; FOXF2; PTEN; GSK3B; TIAM1
	GO:0001952	Regulation of cell-matrix adhesion	6/179	1.48× 10^−2^	BCL2; RCC2; PTEN; GSK3B; THBS1; ACER2
	GO:0060317	Cardiac epithelial to mesenchymal transition	3/179	4.00 × 10^−2^	PDCD4; SNAI2; SMAD4
	GO:0010810	Regulation of cell-substrate adhesion	7/179	4.20 × 10^−2^	BCL2; RCC2; ACTN4; PTEN; GSK3B; THBS1; ACER2
	GO:0007160	Cell-matrix adhesion	7/179	4.65 × 10^−2^	BCL2; RCC2; PTEN; GSK3B; THBS1; ACER2; TIAM1
MSigDB:H-HALLMARK	Hallmark_ Epithelial_ Mesenchymal _Transition	Hallmark_ Epithelial_ Mesenchymal _Transition	9/179	1.65 × 10^−2^	NTM; SNAI2; PCOLCE2; BDNF; CADM1; NOTCH2; THBS1; SERPINH1; LOX

EMT = Epithelial–to–mesenchymal transition; KEGG = Kyoto Encyclopedia of Genes and Genomes; GO-BP = Gene Ontology-Biological Process; MSigDB:H = Molecular Signatures Database: Hallmark. ^1^ Adjusted *p*-value for multiple hypothesis correction used Benjamini and Hochberg procedure.

## Data Availability

The genotypic and phenotypic data for Prostate adenocarcinoma (PRAD) cohort are available at The Cancer Genome Atlas (TCGA) portal (http://portal.gdc.cancer.gov/projects). Serum miR-182 expression data is available at Gene Expression Omnibus (GEO) portalhttp://www.ncbi.nlm.nih.gov/geo/ (accessed on 10 May 2022); datasets GSE112264 and GSE113486. Analysis tools are listed in Methods, and other datasets analyzed in the present study are available from the published papers that have been cited in this manuscript.

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
