# Peer review of "MiR-182 Is Upregulated in Prostate Cancer and Contributes to Tumor Progression by Targeting MITF"

_ijms, 2023, doi:10.3390/ijms24031824_

Round 1
Reviewer 1 Report
MicroRNAs (miRNAs) are known to play a role in regulating the expression of genes related to the formation and development of cancers, including prostate cancer. However, the multitude of cellular pathways in which miRNAs are involved means that there is still a lack of information that could contribute to their use as diagnostic markers or therapeutic molecules. Therefore, any work leading to a detailed understanding of how gene expression is modulated by these non-coding RNAs is very much needed. This article fits this topic.
The article is very well prepared, the research was designed appropriate and methods clearly presented. However I suggest that in the section Cell culture and transfections – the information about density of each cell culture used for transfection research is needed. This part should be described in more detail or have references to the literature. There are no information about the sequence of miR-182 precursor or negative control (how was it constructed).
I consider the presented work valuable and after few corrections, I think that it should be published in International Journal of Molecular Sciences. The connection between miR-182 and MITF has been documented on many levels, I found the article very valuable.
Author Response
MicroRNAs (miRNAs) are known to play a role in regulating the expression of genes related to the formation and development of cancers, including prostate cancer. However, the multitude of cellular pathways in which miRNAs are involved means that there is still a lack of information that could contribute to their use as diagnostic markers or therapeutic molecules. Therefore, any work leading to a detailed understanding of how gene expression is modulated by these non-coding RNAs is very much needed. This article fits this topic.
The article is very well prepared, the research was designed appropriate and methods clearly presented.
We are pleased the reviewer finds the work to be important, well-written and suitably conducted.
However I suggest that in the section Cell culture and transfections – the information about density of each cell culture used for transfection research is needed. This part should be described in more detail or have references to the literature. There are no information about the sequence of miR-182 precursor or negative control (how was it constructed).
The methods section has now been amended to include this information (lines 91-99). Please note the miR-182 mimic and inhibitor molecules are commercially available, so more information is available at website of manufacturer ThermoFisher Scientific https://www.fishersci.co.uk/shop/products/i-mir-i-vana-mirna-mimic-13/13426696
I consider the presented work valuable and after few corrections, I think that it should be published in International Journal of Molecular Sciences. The connection between miR-182 and MITF has been documented on many levels, I found the article very valuable.
We are pleased the reviewer finds the work valuable and worthy of publication in IJMS.
Reviewer 2 Report
This study qualified Mir-182 as a functional biomarker in prostate cancer. The Mir-182 target, MITF, is reduced in cancer versus normal tissue. Other targets of Mir-182 are genes associated with EMT. In a K-M analysis, high Mir-182 is associated with reduced diseased-free survival and progression-free survival.
Overall, the data are solid and well represented and the paper is well written and organized, however that are several major questions and concerns that diminish the enthusiasm of this reviewer:
1. the MITF gene is not listed in Table 1 and 2. Although the gene is proposed by the authors to be associated with EMT, it does not appear in an enrichment analysis of miR-182 associated genes. Please explain why this is the cases. Is MITF not included in the gene list, or is the mRNA not associated with miR-182 in this particular type of statistical analysis.
2. Please move Figure 1S to the main paper. It seems that miR-182 regulated genes are mostly enriched in the “prostate cancer” gene set. What are the genes that drive the enrichment for prostate cancer and are those androgen-regulated genes ? Is there a correlation between miR-182 and androgen-regulated cistrome ?
3. MITF is expressed in cells with neuronal differentiation. In Figure 1S, a pathway analysis of miR-128 regulated genes reveal pathways related to prostate cancer, such as FOXO, and in addition pathways that are related with aggressive prostate cancer, such as p53, proliferation and AKT. The authors should consider an association of miR-182 with cell plasticity, which encompasses both, neuronal/neuroendocrine and EMT.
4. In the protein atlas, it appears that MITF protein is expressed in histiocytes (a subtype of macrophages), in both prostate and lung. Histiocytes may also be prominent in cancer samples. The authors should exclude a contribution of histocytes and if necessary adjust for histiocytes when analyzing MITF mRNA expression.
5. The novelty of the study solely relies on the demonstration that a known miR-182 – MITF exists in prostate cancer. The authors state that the miR-182 – MITF axis has been shown in other cancer types, but not yet in prostate cancer.
6. the authors do not consider publications that MITF can be pro-oncogenic in uveal melanoma PMID: 35682684, gastric cancer PMID: 31171711 etc. The activity of MITF seems to be context dependent and both, high and low expression of MITF can lead to functionalities, such as invasion associated with low MITF and proliferation associated with high MITF that are pro-oncogenic PMID: 25538895
Author Response
REVIEWER 2
This study qualified Mir-182 as a functional biomarker in prostate cancer. The Mir-182 target, MITF, is reduced in cancer versus normal tissue. Other targets of Mir-182 are genes associated with EMT. In a K-M analysis, high Mir-182 is associated with reduced diseased-free survival and progression-free survival.
Overall, the data are solid and well represented and the paper is well written and organized, however that are several major questions and concerns that diminish the enthusiasm of this reviewer:
We are pleased the reviewer finds the work to be well-presented and we endeavour below to address the concerns noted
- the MITF gene is not listed in Table 1 and 2. Although the gene is proposed by the authors to be associated with EMT, it does not appear in an enrichment analysis of miR-182 associated genes. Please explain why this is the cases. Is MITF not included in the gene list, or is the mRNA not associated with miR-182 in this particular type of statistical analysis.
We were not surprised that MITF did not appear in these lists, as they represent selected results showing prostate- and EMT-associated processes linked with miR-182. Since little published evidence exists to link miR-182/MITF with prostate cancer or EMT, it wasn’t expected that MITF would appear against the terms listed. Indeed, this was part of the impetus for our work here, as we felt there was scope for novel findings to link miR-182, MITF and EMT in prostate cancer.
As the reviewer astutely notes, this is also likely to be due to the type of statistical analysis performed, which has cut-off thresholds that may be more or less stringent, dependent on the settings.
However, we would reassure the reviewer that MITF certainly featured in the functional analysis overall. The processes that do feature MITF included many related to melanoma, which was expected given the obvious links of MITF in this disease, as well as some other cancers (e.g. Renal, uveal).
Importantly, while the functional enrichment analysis is a useful ‘screening’ tool, it is always important to review the literature independently for evidence also. In the case of MITF, we did indeed find studies linking it to EMT, which corroborated our own data (Figure 4 in revised manuscript). Therefore, we were happy to pursue MITF as a possible EMT-linked target.
- Please move Figure 1S to the main paper. It seems that miR-182 regulated genes are mostly enriched in the “prostate cancer” gene set. What are the genes that drive the enrichment for prostate cancer and are those androgen-regulated genes ? Is there a correlation between miR-182 and androgen-regulated cistrome?
We have moved the figure to the paper (Figure 3 in revised manuscript) and renumbered the manuscript figures accordingly. The genes that drive the enrichment are those listed in Table 1. As the reviewer will note, these do indeed include androgen-regulated genes and/or genes involved in pathways related to androgen receptor signalling (e.g. GSK3B, PTEN, FOXO…). This does implicate miR-182 in the androgen-regulated cistrome and we have already referenced studies that have proposed this (Souza et al, 2022; Wang et al, 2018). We have now added sentences to the discussion (lines 385-389) to emphasise this aspect more, as well as citing a new reference which specifically links miR-182 with androgen receptor regulation (Yao et al, 2016; Ref 53 in Revised manuscript).
- MITF is expressed in cells with neuronal differentiation. In Figure 1S, a pathway analysis of miR-128 regulated genes reveal pathways related to prostate cancer, such as FOXO, and in addition pathways that are related with aggressive prostate cancer, such as p53, proliferation and AKT. The authors should consider an association of miR-182 with cell plasticity, which encompasses both, neuronal/neuroendocrine and EMT.
The reviewer raises an important point that provides further biological rationale for a link between miR-182 and EMT. In light of this, we have now included sentences (lines 257-260 and 408-410) and an extra citation for a comprehensive review paper of MITF function (Goding & Arnheiter, 2019; Ref 42 in revised manuscript), including its contribution to pathways leading to cancer (see also response to Point 6 below). We have also added sentences to specifically consider the possibility of MITF (and miR-182) contributing to EMT-induced cellular plasticity in neuroendocrine prostate cancer (lines 419-424), again supported by two new citations (Cham et al, 2021 and Sreekumar & Saini, 2022; Refs 61 and 62 in revised manuscript).
- In the protein atlas, it appears that MITF protein is expressed in histiocytes (a subtype of macrophages), in both prostate and lung. Histiocytes may also be prominent in cancer samples. The authors should exclude a contribution of histocytes and if necessary adjust for histiocytes when analyzing MITF mRNA expression.
The data we are analysing in this paper does not give us the opportunity to select or exclude specific cell-types. Note also that the TCGA analyses were based on genomic, rather than proteomic data. However, the reviewer raises an important point as tumor heterogeneity is a major issue, both in the interpretation of data and in the extrapolation to clinical usefulness. With that in mind, we have added sentences to acknowledge this issue (lines 424-432) supported by an extra citation (Haffner et al, 2021; Ref 63 in revised manuscript), to highlight the challenges. We also state that expression of MITF and/or miR-181 may well be dependent on certain cell-types within the tumor. We propose the single-cell analysis and/or focus on disease sub-types as prudent approaches to unravel their combined biological function of MITF/miR-182.
- The novelty of the study solely relies on the demonstration that a known miR-182 – MITF exists in prostate cancer. The authors state that the miR-182 – MITF axis has been shown in other cancer types, but not yet in prostate cancer.
The reviewer is correct to state the main novelty of the study is that it is the first to show the miR-182/MITF interaction in prostate cancer. Furthermore, to our knowledge, this is also the first to show this correlation in human cancer datasets in silico, and only the third paper published showing in vitro evidence that MITF is a target of miR-182. We have added text to emphasise this novelty (lines 260-265). We submit that publishing this work will help us and others discover further novel mechanisms that might underpin this association. As it stands, I am sure the reviewer recognises our wish to publish our findings to date now.
- the authors do not consider publications that MITF can be pro-oncogenic in uveal melanoma, gastric cancer PMID: 31171711 etc. The activity of MITF seems to be context dependent and both, high and low expression of MITF can lead to functionalities, such as invasion associated with low MITF and proliferation associated with high MITF that are pro-oncogenic PMID: 25538895
In the interests of space, we did not cite every study that looked at the link between MITF and cancer. However, we did cite one of the references the reviewer suggests above (PMID: 35682684; Gelmi et al, 2022, ref 41 in original submission], using it to demonstrate the link between MITF and EMT in melanoma.
Nevertheless, the reviewer comment has prompted us to further emphasise the link between MITF and various cancers, so we have add sentences (lines 257-260 and 408-410) and cited a useful review paper which evaluates a range of studies on this (Goding & Arnheiter, 2019; Ref 42 in revised submission). This also helps evidence that MITF may play a dual role and that this activity is tissue dependent. (By extension, regulation of MITF by miR-182 is an important mechanism by which its expression may be altered, either by repressing, or allowing over-expression to occur.)
Round 2
Reviewer 2 Report
The authors address all the concerns of this reviewer. They could do an analysis to deconvolute the cell types in the samples using RNA data in TCGA and determine whether MITF expression correlates with specific immune cell subtypes. Instead of a data analysis, the authors addressed the point as a limitation of the study in the Discussion section. This is perhaps not optimal, but acceptable to this reviewer.
Author Response
The authors address all the concerns of this reviewer. They could do an analysis to deconvolute the cell types in the samples using RNA data in TCGA and determine whether MITF expression correlates with specific immune cell subtypes. Instead of a data analysis, the authors addressed the point as a limitation of the study in the Discussion section. This is perhaps not optimal, but acceptable to this reviewer.
We are pleased the reviewer finds the revision acceptable, but we thought it prudent to add some extra analyses to address the likelihood of MITF involvement in immune response. WE know from looking at Protein Atlas that macrophages are one of the most common cells in prostate tumor biopsies. We therefore have added a supplementary figure (Figure S6) showing that MITF shows significant positive correlation with several markers of tumor-associated macrophages, which are known to play prominent role in progressing prostate tumor growth. We have added text to the manuscript to discuss this (Lines 430-433), supported by a new citation (Masetti et al, 2022; Ref 64 in clean manuscript).
Again, we would thank the reviewer for their time and critical evaluation, which have helped improve the manuscript.